# Individual and community-level determinants of skilled birth attendant delivery in Ethiopia; multilevel analysis

Hiwotie Getaneh Ayalew[1]*, Kibir Temesgen Assefa[1], Selam Yibeltal Desalegn[1], Tiruye Tilahun Mesele[2], Tazeb Alemu Anteneh[2], Nebiyu Solomon Tibebu[2], Alemneh Mekuriaw Liyew[3]

1 Department of Midwifery, School of Nursing and Midwifery, College of Medicine and Health Sciences, Wollo University, Dessie, Ethiopia, 2 Department of Midwifery, School of Midwifery, College of Medicine and Health Sciences, University of Gondar, Gondar, Ethiopia, 3 Department of Epidemiology and Biostatistics, Institute of Public Health, College of Medicine and Health Sciences, University of Gondar, Gondar, Ethiopia

* hiwotiegeta27@gmail.com

## Abstract

### Introduction

Skilled birth attendant (SBA) delivery is defined as assisting birth by a trained healthcare provider, which is vital for the health of mothers and newborns. Improving maternal health is one of the world health organization's (WHO) key priorities and skilled birth attendant delivery is one of the four pillars of the initiative for safe motherhood to reduce maternal mortality. Therefore, this study aimed to assess the individual and community-level factors associated with SBA delivery in Ethiopia.

### Method

A secondary data analysis was conducted using the 2019 Mini Ethiopian demographic and health survey. A total of 5,527 (weighted) live births were included in the analysis. A multilevel logistic regression model was fitted using Stata 14.0 to identify individual and community-level factors associated with SBA delivery. Finally, AOR with 95% CI and random effects were reported.

### Result

In this study after fitting a multilevel model, women with poor (AOR = 0.44 95%CI 0.32–0.61) and middle wealth index (AOR = 0.64;95% CI 0.46–0.87), multipara (AOR = 0.39;95% CI 0.28–0.55) and grand multipara (AOR = 0.46;95% CI 0.29–0.72), women from rural areas (AOR = 0.34;95% CI 0.16–0.72) and high community poverty level (AOR = 0.40;95% CI 0.21–0.76) had decreased odds of having SBA delivery. Whereas those who initiated Antenatal care (ANC) visits in the first trimester (AOR = 2.65; 95% CI 1.52–4.65) and second trimester (AOR = 1.87:95%CI 1.09–3.20) had increased odds of having SBA delivery in Ethiopia.

**Data Availability Statement:** All the data used in this study are publicly available at www.dhsprogram.com. Detaills are within the paper.

**Funding:** The authors received no specific funding for this work.

**Competing interests:** All authors declare that they have no competing interests exist.

**Abbreviations:** ANC, Antenatal Care; AOR, Adjusted odds Ratio; CSA, Central Statistical Agency; EDHS, Ethiopian Demographic and Health Survey; ICC, Intra-cluster Correlation Coefficient; ICF, International Classification of Functioning; LLR, log-likelihood Ratio; MEDHS, Mini Ethiopian Demographic and Health Survey; PCV, Proportional Change in Variance; PHC, Population and Housing Census; SBA, Skill Birth Attendant; SDG, Sustainable Development Goal; WHO, World Health Organization.

## Conclusion

In this study socioeconomic factors like wealth index, parity, the timing of ANC visits, place of residency, and community poverty level were significantly associated with SBA delivery. Therefore, it is better to increase timely ANC initiation particularly for women with low levels of income to improve skilled birth attendant delivery.

## Introduction

Skilled birth attendant (SBA) delivery is defined as assisting birth by a trained healthcare provider, which is vital for the health of mothers and newborns since most maternal and newborn deaths occur at the time of childbirth or immediately after birth [1]. Improving maternal health is one of the World Health Organization's (WHO's) key priorities and SBA delivery is of the four pillars of the initiative for safe motherhood to reduce maternal mortality since it offers an opportunity to organize the necessary services for ensuring a healthy mother and newborn [2].

Although a pronounced reduction in maternal mortality was achieved (from 385 in 1990 to 216 deaths per 100,000 live births in 2015) worldwide, sub-Saharan Africa and South Asia are disproportionately affected which accounted for more than three-fourths of these global maternal deaths [3]. In our study setting Ethiopia, although a slight decrease in maternal mortality is observed in the last two decades, still we lost 412 mothers per 100,000 live births as it was reported in the recent national demographic and health survey [4].

Ambitiously Sustainable Development Goal (SDG) has set a target of reducing the maternal mortality ratio to 70 deaths per 100,000 live births by 2030 through strategic interventions of which the SBA delivery service [5]. Despite the WHO's recommendation of every birth to be attended by a skilled health professional regardless of whether the birth occurs in a health facility or at home, still, millions of births occur without the assistance of SBA personnel annually [6].

Moreover, the coverage of SBA delivery in sub-Saharan Africa is not at the expected level with great disparity across different geographic regions ranging from 38.5% in Nigeria to 77% in Kenya [7–10]. Similarly, in Ethiopia, the national coverage of SBA delivery as of 2019 was 50% [4] and lower coverages are reported in different local level studies [11].

Different sociodemographic and socioeconomic factors such as the age of the mother [10, 12, 13], maternal education [13], parity [10, 13], wealth index [14, 15], number of ANC visits [11, 16], marital status [17], place of residence [18–20], knowledge on danger signs of pregnancy [21, 22] and media exposure [23] were some of the driving factors of SBA delivery reported in previous studies. However, apart from the individual maternal factors, community contexts where a woman is dwelling could have a potential effect on obtaining the SBA delivery service which is yet to be investigated in Ethiopia. It is also critical to provide updated evidence on those modified and persistent maternal factors that contribute to the failure to improve coverage of SBA delivery in Ethiopia by using the most recently available national data which might give a clue to policymakers, program designers, and health professionals to take targeted and integrated interventions. Therefore, this study aimed to assess the individual and community-level determinants of SBA delivery in Ethiopia.

## Method

### Study setting

Ethiopia is found in the Horn of Africa and lies between the Equator and Tropic of Cancer, between the 30 N and 150N Latitude or 330 E and 480 E Longitude. The country occupies an

area of approximately 1,127,127 square km. This land area is 1,119,683 square km and the area occupied by water bodies is 7,444 sq. km. Ethiopia is a country rich in geographical diversity. The highest altitude is at Ras Dejen (4,620 m above sea level) and the lowest altitude is at Kobar Sink (120 m below sea level).

Ethiopia is the 13th in the world and the 2nd in Africa's most populous country having about 112 million populations. The majority of the population lives in rural areas. Ethiopia has 3 tiers of health systems such as primary health care unit, secondary health care unit, and tertiary health care unit. Ethiopia is an agrarian country and agriculture accounts for 43 percent of the gross domestic product (GDP) and 84% of the population lives in rural areas. More than 80 percent of the country's total population lives in the regional states of Amhara, Oromia, and SNNP [24].

## Data source, sampling technique, and sample size

Secondary data analysis was done based on the mini Ethiopian demographic and health survey 2019 (MEDHS), which was collected based on a crossectional study design from March 21, 2019, to June 28, 2019. The 2019 MEDHS is a nationally representative cross-sectional survey that was conducted by the Central Statistical Agency (CSA) with assistance from the International Classification of Functioning, Disability, and Health (ICF International).

In this survey, two-stage stratified cluster sampling was used by taking enumeration areas as primary sampling units and households as secondary sampling units. In the first stage, regions were stratified into urban and rural areas. Enumeration areas proportional to the size of the national population were selected from rural and urban clusters. The sample included 305 enumeration areas, 93 in urban and 212 in rural areas [4]. In the second stage, a fixed number of 30 households per cluster were selected with an equal probability of systematic selection from the newly created household listing. All live births from the selected households or visitors who stayed in the household the night before the survey were eligible to be interviewed. For this study, data were extracted from kids' record datasets. Participants with missed outcome variables were excluded and a total of 5,527 (weighted) live births five years preceding the survey were included in this study.

## Study variables

**Dependent variable.** The outcome variable of this study was SBA delivery, which was defined as live births delivered with the assistance of doctors, nurses, midwives, health officers, and health extension workers [4]. The SBA deliveries were recoded as "1" otherwise "0" to fit the model.

**Independent variables.** In this study, both individual and community-level factors were considered independent variables. The individual-level factors included were the educational status of the mother (no formal education, primary, secondary, higher), marital status (married, unmarried), maternal age (15–24, 25–34, and 35–49), parity (primipara, multipara, grand multipara), wealth status (poor, middle, rich), Time of ANC visit (first trimester, second trimester, third trimester), Number of ANC visit (No visit,1–4 visits, greater than 4 visits), whereas the community-level factors included were the place of residency (urban, rural) and region with three categories such as larger central (Tigray, Amhara, Oromia, and Sothern Nations Nationalities and Peoples Region), small peripherals (Afar, Somali, Benishangul, and Gambella), and metropolis (Harari, Dire Dawa, and Addis Ababa). The community-level factors, like community media exposure, were obtained by aggregating the individual level media exposure in each cluster by using the proportion of those who had media exposure and this community-level media exposure shows the overall media exposure in the community.

Median values were used to categorize as high and low because the aggregated variable had a skewed distribution. Community poverty level was also obtained by an aggregated proportion of the poor and shows the overall poverty status of the community within the specific cluster. It was again categorized as high and low community poverty levels based on the median value due to the skewed nature of proportion.

**Data management and analysis.**   The data was accessed from the DHS program's official database after permission was granted through an online request by explaining and writing an abstract about the objective of the study [25].

The outcome variable with important factors was extracted from the mini Ethiopian Demographic and Health Survey 2019, using a kid's record data set. Labeling, recording, and analysis were done using STATA 14. The data were analyzed using sampling weight, primary sampling unit, and strata before any statistical analysis to restore the representativeness of the survey for unequal sample sizes across clusters and to tell the STATA to take into account the sampling design when calculating standard errors to get reliable statistical estimates. Basic descriptive statistics were used to show the distribution of eligible participants, and sociodemographic, individual, and community-level characteristics.

**Multilevel logistic regression analysis.**   The outcome variable was dichotomized as skilled and unskilled birth attendant delivery for analysis. Because of the hierarchical nature of the data and the dichotomous outcome variable, the multilevel logistic regression model was used. Before fitting the multilevel logistic regression model for individual and community-level variables the chi-square assumption was checked. variables that were statistically significant in the Bivariable multilevel logistic regression model at a p-value less than 0.2 were considered for multivariable analysis.

Four models were fitted. The first was the null model containing no exposure variables which was used to check variation in the community and provide evidence to assess random effects at the community level. The second model was the multivariable model adjustment for individual-level variables and model three was adjusted for community-level factors. In the fourth model, both individual and community-level variables were fitted with the outcome variable.

Regarding the measures of variation (random effects) intracluster correlation coefficient (ICC) and Proportional Change in Variance (PCV) were used. The ICC (Intracluster correlation coefficient) quantifies the degree of heterogeneity of skilled birth attendant delivery between clusters (enumeration areas) in Ethiopia.

$$\text{ICC} = \delta 2/(\delta 2 + \pi 2/3)$$

Where: - $\delta^2$-between cluster variance, $\pi^2/3$ -within-cluster variance

Whereas, PCV measures the total variation attributed by individual-level factors and community-level factors in the multilevel model as compared to the null model which was computed by using the following formula; PCV = (variance in null model-variance in the full model)/ variance null model [26]. Since the models were hierarchical and nested, the model comparison was done using deviance and Log likelihood ratio. Consequently, the best-fitted model with the highest Log likelihood ratio and the lowest deviance to the data was used. Finally, the adjusted odds ratio with a 95% confidence interval at a p-value less than 0.05 was used to declare statistical significance.

**Ethical approval and consent to participate.**   This study was a secondary analysis of the 2019 Ethiopian Demographic and Health Survey data. Ethical clearance was obtained from measure DHS after submitting a proposal to DHS Program and permission was confirmed from the International Review Board of Demographic and Health Surveys (DHS) program

data archivists to download the dataset for this study. The data used in this study are publicly available, aggregated secondary data that hasn't any personal identifying information that can be linked to study participants. The confidentiality of data was maintained anonymously during data collection of the MEDHS 2019. No formal ethical approval was required in this particular study. Information obtained from the data set was disclosed to any third person.

## Result

### Individual and community-level characteristics of the study participants

Among a total of 5,527 participants, 2,962 (53.58%) participants had no formal education. The mean age of study subjects was 28.64 with a standard deviation of 6.48 and most of the respondents 5,225 (94.53%) were married. Regarding the wealth index, 2,518 (45.56%) participants were rich.

A total of 305 communities (clusters) were included in the study. Of 5,527 participants 4,160 (75.27%) lived in the rural community and 4,738 (89.58%) lived in large central regions. Above half of the participants, 3,361 (60.80%) lived in communities with low poverty levels and 3,009 (54.44%) lived in communities with high community media exposure (Table 1).

### Random effect analysis results

In the null model, variance component analysis was performed to decompose the total variance of SBA delivery between clusters. The cluster-level variance which indicates the total variance of SBA delivery that can be attributed to the context of the community in which the mothers were living was estimated. The applicability of the multi-level logistic regression model in the analysis was justified by the significance of the community-level variance [community variance = 4.18; standard error (SE) = 0.54; P-value, 0.00], indicating the existence of significant differences between communities regarding SBA delivery. The community variance was expressed as an intracluster correlation coefficient (ICC). The ICC was 55.97%, which revealed that 55.97% of the total variance of SBA in Ethiopia could be attributed to the context of the communities (clusters) where the mothers were living. In the full model community variance (community variance = 4.18; SE 0.54; P-value, 0.00) remained significant even after considering some contextual risk factors for SBA delivery. The PCV in this model was 54.54%. It suggested that 54.54% of community variance observed in the null model was explained by both community and individual-level variables.

### Model comparison

The combined multilevel logistic regression model (model IV) which has a high log-likelihood ratio (-1359.73) and low (2719.46) deviance was the best-fitted model to data (Table 2).

### Fixed effect analysis results

In the bi-variable multilevel logistic regression analysis marital status, wealth index, maternal age, mother's education, parity, time of ANC visit, number of ANC visits, place of residence, region, community poverty level, community media exposure were significant at p-value<0.2 and fitted for Multi-variable analysis. Multivariable multilevel logistic regression analysis was fitted to identify determinants of SBA delivery in Ethiopia. In the final model (Model IV) wealth index, parity, time of ANC visit, place of residency, and community poverty were significantly associated with SBA delivery.

After keeping another individual, community-level factors and random effect constant the odds of having SBA among poor women were 0.44 (AOR = 0.44;95% CI 0.32–0.61) and middle

**Table 1. Background characteristics of study participants, MEDHS 2019 (N = 5,527 weighted).**

| Individual level variables | Frequency (weighted) | Percentage |
|---|---|---|
| Maternal age (years) | | |
| 15–24 | 1,295 | 23.42 |
| 25–34 | 2,949 | 53.36 |
| 35–49 | 1,283 | 23.22 |
| Maternal education | | |
| No formal education | 2,962 | 53.58 |
| Primary education | 1,956 | 35.40 |
| Secondary education | 415 | 7.51 |
| Higher education | 192 | 3.51 |
| Marital status | | |
| Married | 5,225 | 94.53 |
| Unmarried | 302 | 5.47 |
| Wealth index | | |
| Poor | 2,518 | 45.56 |
| Middle | 1,044 | 18.88 |
| Rich | 1,965 | 35.55 |
| Parity | | |
| Primiparous | 825 | 14.93 |
| Multiparous | 3,148 | 56.95 |
| Grand multiparous | 1,554 | 28.12 |
| Time of ANC visit | | |
| First trimester | 1,092 | 37.37 |
| Second trimester | 1,721 | 58.87 |
| Third trimester | 110 | 3.76 |
| Number of ANC visits | | |
| No visit | 1,013 | 25.81 |
| 1–4 visits | 2,145 | 54.62 |
| >4 visits | 769 | 19.57 |
| **Community-level variables** | | |
| Place of residency | | |
| Urban | 1,367 | 24.73 |
| Rural | 4,160 | 75.27 |
| Region | | |
| Large central | 4,738 | 85.73 |
| Small peripheral | 586 | 10.61 |
| Metropolis | 202 | 3.66 |
| Community poverty | | |
| Low | 3,361 | 60.80 |
| High | 2,166 | 39.20 |
| Community media exposure | | |
| Low | 2,518 | 45.56 |
| High | 3,009 | 54.44 |

wealth index 0.63 (AOR = 0.63;95% CI 0.46–0.87) were decreased by 56% and 37% respectively as compared to women in rich wealth index. The odds of having SBA delivery among multipara 0.39 (AOR = 0.39; 95%CI 0.28–0.55) and grand multipara women 0.46 (AOR = 0.46; 95% CI 0.29–0.72) was decreased by 61% and 56% respectively as compared to primipara women.

**Table 2. Random effects and model fitness.**

| Random effects | Model l | Model ll | Model lll | Model lV |
|---|---|---|---|---|
| Community variance(SE) | 4.18(0.54) | 2.04(0.33) | 2.14(0.28) | 1.90(0.31) |
| ICC (%) | 55.97 (0.49–0.62) | 38.36(0.31–0.46) | 39.41(0.33–0.45) | 36.71(0.29–0.44) |
| PCV (%) | Reference | 51.19 | 48.80 | 54.54 |
| **Model fitness** | **Model I** | **Model ll** | **Model lll** | **Model lV** |
| Log-likelihood | -2807.56 | -1385.52 | -2722.45 | -1359.73 |
| Deviance(-2LLR) | 5615.12 | 2771.04 | 5444.90 | 2719.46 |

ICC: intracluster correlation coefficient; PCV: proportional change in variance; SE: standard error; LLR: log-likelihood ratio

Women who started ANC visit first trimester and second trimester had 1.87 (AOR = 1.87:95% CI 1.09–3.20) and 2.65 (AOR = 2.65;95% CI 1.52–4.65) respectively times higher odds of having SBA delivery as compared to women who initiated the first ANC visit at the third trimester. The community-level factors have also shown a significant impact on SBA delivery. For instance, the likelihood of having SBA delivery among rural women decreased by 66% (AOR = 0.34; 95% CI 0.16–0.72) as compared to women who lived in urban areas. Similarly, the odds of having SBA delivery among women who live in high community poverty levels were decreased by 60% (AOR = 0.40;95% CI 0.21–0.76) as compared to women who lived in low community poverty levels (Table 3).

## Discussion

This study investigated individual and community-level determinants of skilled birth attendant delivery in Ethiopia. The maternal wealth index was significantly associated with SBA delivery where women with poor and middle wealth index had lower odds of SBA delivery as compared to women with rich wealth index. This study finding was supported by studies done in Afghanistan [14], India [20], East Africa [10], and Ethiopia [18]. This might be due to that women's income affects maternal healthcare service utilization since women with low economic status face difficulty in covering transportation and some medical-related costs [27].

Parity was also another factor that was associated with SBA delivery in Ethiopia. The odds of having SBA delivery among multipara and grand multipara women were decreased by 61% and 56% respectively as compared to primipara women. This study finding was supported by studies done in Bangladesh [28], Ghana [13], and East Africa [10]. The possible reason might be that primigravidas are new to pregnancy, labor, and delivery which might push them to give birth at the health institution. Consequently, they might have the chance to get trained healthcare providers as compared to their counterparts. On the other hand the multipara and grand multipara, women might have less anxiety and more confidence in labor which might in turn lead to home delivery [18, 29].

Additionally, the timing of the ANC visit was significantly associated with SBA delivery where women who started ANC visit in the first trimester and the second trimester had more than nearly two times and three times higher odds of having SBA delivery respectively as compared to women who started in the third trimester. This study finding was concordant with studies done in Ethiopia [27]. Possibly it might be due to women with early ANC visits have higher satisfaction with the care quality and hence being more likely to use health services for delivery. It might also be because timely ANC visits expose women to more health education and counseling which helps to increase service utilization [18, 27].

Our study has also highlighted that the community setting where women have been dwelling has a significant effect on the utilization of SBA delivery service. For example, the

**Table 3. Multilevel logistic regression analysis of individual and community-level determinants of SBA delivery in Ethiopia, MEDHS 2019.**

| Characteristics | Model I | Model II | Model lll | Model IV |
|---|---|---|---|---|
| Fixed effect | | AOR(95%CI) | AOR(95%CI) | AOR(95%CI) |
| Maternal age (years) | | | | |
| 15–24 | - | 1.00 | - | 1.00 |
| 25–34 | - | 0.95(0.70–1.31) | - | 0.89(0.65–1.22) |
| 35–49 | - | 1.33(0.88–1.99) | - | 1.79(0.81–1.83) |
| Maternal education | | | | |
| No | - | 0.44(0.21–0.90) | - | 0.53(0.25–1.11) |
| Primary | - | 0.71(0.35–1.45) | - | 0.86(0.41–1.77) |
| Secondary | - | 1.71(0.56–2.73) | - | 1.47(0.65–3.29) |
| Higher | - | 1.00 | - | 1.00 |
| Marital status | | | | |
| Married | - | 1.00 | - | 1.00 |
| Unmarried | - | 0.95(0.59–1.53) | - | 0.87(0.54–1.42) |
| Wealth index | | | | |
| Poor | - | 0.36 (0.26–0.50) | - | 0.44(0.32–0.61)* |
| Middle | - | 0.55 (0.40–0.77) | - | 0.63(0.46–0.87)* |
| Rich | - | 1.00 | - | - |
| Parity | | | | |
| Primipara | - | 1.00 | - | 1.00 |
| Multipara | - | 0.38(0.27–0.53) | - | 0.39(0.28–0.55)* |
| Grand multipara | - | 0.43(0.27–0.68) | - | 0.46(0.29–0.72)* |
| Time of ANC visit | | | | |
| First trimester | - | 2.66(1.54–4.64) | - | 2.65(1.52–4.65)* |
| Second trimester | - | 1.82 (1.06–3.10) | - | 1.87(1.09–3.20)* |
| Third trimester | | 1.00 | | 1.00 |
| Number of ANC visit | | | | |
| No visit | | 0.98(0.13–7.16) | | 1.02(0.13–8.06) |
| 1–4 visits | | 0.81(0.6214–1.06) | | 0.84(0.64–1.09) |
| >4 visits | | 1.00 | | 1.00 |
| Residency | | | | |
| Urban | - | - | 1.00 | 1.00 |
| Rural | - | - | 0.19(0.09–0.36) | 0.34(0.16–0.72)* |
| Region | | | | |
| Large central | - | - | 0.53(0.23–1.22) | 1.27(0.08–2.86) |
| Small peripheral | - | - | 0,28(0.11–0.69) | 0.36(0.10–1.27) |
| Metropolis | - | - | 1.00 | 1.00 |
| Community poverty | | | | |
| Low | - | - | 1.00 | 1.00 |
| High | - | - | 0.24(0.13–0.43) | 0.40(0.21–0.76)* |
| Community media exposure | | | | |
| Low | - | - | 0.49(0.28–0.86) | 0.79(0.43–1.43) |
| High | - | - | 1.00 | 1.00 |

*Significant, 1.00: reference

likelihood of SBA delivery among women living in rural communities decreased by 66% as compared to women who lived in urban communities. This study finding was in line with studies done in Ethiopia [17, 21, 27] and India [20]. The possible explanation might be that

women living in urban areas have more access to health information, and access to nearby services compared to rural areas [11, 18, 30]. Similarly, women who lived in high community poverty settings had lower odds of SBA delivery as compared to women who lived in a community with low poverty. This finding was supported by a study done in Ethiopia [31]. The explanation might be mothers who live in high-poverty communities suffer to overcome their basic needs including health care and living costs. In addition to this, women from the poor community might lack access to delivery care services due to limited infrastructures in their specific locality which is common in resource-limited countries like Ethiopia [32].

The strength of this study was it used a large data set representing the whole country and applied sampling weight to make it nationally representative and to produce a reliable estimate. This study also used a multilevel model for analysis, which gives unbiased standard errors. Whereas, the limitation of this study could be the possibility of committing social desirability and recall bias since it was a community-based survey.

## Conclusion

In this study determinants like wealth index, parity, the timing of ANC visits, place of residency, and community poverty were significantly associated with SBA delivery in Ethiopia. It is better to improve timely ANC initiation particularly for women with low levels of income, and improve maternal and reproductive health service utilization by improving women's income and community living standards.

## Acknowledgments

We would like to thank the measure DHS program for their permission to access the 2019 MEDHS datasets.

## Availability of data and materials

All the data used in this study are publicly available at www.dhsprogrwm.com without restriction.

## Author Contributions

**Conceptualization:** Hiwotie Getaneh Ayalew, Kibir Temesgen Assefa, Selam Yibeltal Desalegn, Tiruye Tilahun Mesele, Tazeb Alemu Anteneh, Nebiyu Solomon Tibebu, Alemneh Mekuriaw Liyew.

**Data curation:** Hiwotie Getaneh Ayalew, Kibir Temesgen Assefa, Selam Yibeltal Desalegn, Tiruye Tilahun Mesele, Tazeb Alemu Anteneh, Nebiyu Solomon Tibebu, Alemneh Mekuriaw Liyew.

**Formal analysis:** Hiwotie Getaneh Ayalew, Kibir Temesgen Assefa, Selam Yibeltal Desalegn, Tiruye Tilahun Mesele, Tazeb Alemu Anteneh, Nebiyu Solomon Tibebu, Alemneh Mekuriaw Liyew.

**Investigation:** Hiwotie Getaneh Ayalew.

**Methodology:** Hiwotie Getaneh Ayalew, Kibir Temesgen Assefa, Selam Yibeltal Desalegn, Tiruye Tilahun Mesele, Tazeb Alemu Anteneh, Nebiyu Solomon Tibebu, Alemneh Mekuriaw Liyew.

**Software:** Hiwotie Getaneh Ayalew, Kibir Temesgen Assefa, Selam Yibeltal Desalegn, Tiruye Tilahun Mesele, Tazeb Alemu Anteneh, Nebiyu Solomon Tibebu, Alemneh Mekuriaw Liyew.

**Supervision:** Hiwotie Getaneh Ayalew, Kibir Temesgen Assefa, Selam Yibeltal Desalegn, Tiruye Tilahun Mesele, Tazeb Alemu Anteneh, Nebiyu Solomon Tibebu, Alemneh Mekuriaw Liyew.

**Validation:** Hiwotie Getaneh Ayalew, Kibir Temesgen Assefa, Selam Yibeltal Desalegn, Tiruye Tilahun Mesele, Tazeb Alemu Anteneh, Nebiyu Solomon Tibebu.

**Visualization:** Hiwotie Getaneh Ayalew, Kibir Temesgen Assefa, Selam Yibeltal Desalegn, Tiruye Tilahun Mesele, Tazeb Alemu Anteneh, Nebiyu Solomon Tibebu.

**Writing – original draft:** Hiwotie Getaneh Ayalew, Kibir Temesgen Assefa, Selam Yibeltal Desalegn, Tiruye Tilahun Mesele, Tazeb Alemu Anteneh, Nebiyu Solomon Tibebu, Alemneh Mekuriaw Liyew.

**Writing – review & editing:** Hiwotie Getaneh Ayalew, Kibir Temesgen Assefa, Selam Yibeltal Desalegn, Tiruye Tilahun Mesele, Tazeb Alemu Anteneh, Nebiyu Solomon Tibebu, Alemneh Mekuriaw Liyew.

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
