## [Decision Letter · Decision Letter 0]

12 Jan 2023

PONE-D-22-07743INDIVIDUAL AND COMMUNITY-LEVEL DETERMINANTS OF SKILLED BIRTH ATTENDANT DELIVERY IN ETHIOPIA; MULTILEVEL ANALYSISPLOS ONE

Dear Dr. Hiwotie

Thank you for submitting your manuscript to PLOS ONE. After careful consideration, we feel that it has merit but does not fully meet PLOS ONE’s publication criteria as it currently stands. Therefore, we invite you to submit a revised version of the manuscript that addresses the points raised during the review process.

ACADEMIC EDITOR: you are expected to address all comments given to you from both academic editors and reviewers. Be sure to:you have addressed English language edition before submitting again Ensure you have followed the PLOS one journal manuscript submission guideline welland also ensure that you have addressed the comments given for you from the reviewers line by line. its expected from you to address the statistical analysis method again 

We look forward to receiving your revised manuscript.

Kind regards,

Seifadin Ahmed Shallo, MPH

Academic Editor

PLOS ONE

Journal Requirements:

“no”

3.Thank you for stating the following in your Competing Interests section: 

“NO”

Additional Editor Comments (if provided):

strictly follwo the PLOS one journal manuscript submission guideline before submitting your manuscript again

Reviewers' comments:

Reviewer's Responses to Questions

**Comments to the Author**

1. Is the manuscript technically sound, and do the data support the conclusions?

Reviewer #1: Partly

Reviewer #2: Yes

2. Has the statistical analysis been performed appropriately and rigorously? 

Reviewer #1: Yes

Reviewer #2: Yes

3. Have the authors made all data underlying the findings in their manuscript fully available?

Reviewer #1: Yes

Reviewer #2: Yes

4. Is the manuscript presented in an intelligible fashion and written in standard English?

Reviewer #1: No

Reviewer #2: Yes

5. Review Comments to the Author

Reviewer #1: Reviewer Comments

#1. There are numerous typographical and grammatical problems throughout the document which need thorough revision

#2. In the Introduction, Additional justification is needed in the introduction section to justify the novelty of the study, “Even though skilled birth attendant delivery depends on both individual and community-level determinants, still, limited studies have been done beyond individual-level factors” is not enough of a justification since numerous significant factors this study reported had also been reported by other similar studies (which are used in the present study as a reference also)

#3. In the Method section, It is important to specify whether the independent variables (individual and community) were taken as they are or were classified by the authors, for instance, was PCA conducted for the wealth Index?

#4. It is also important to include the measurement section in the Method for some of the variables especially for Community poverty, Community media exposure---

#5. The writing especially in the result section could be improved, what is written is stand-alone sentences, try connecting using conjunctions during interpretations. Additionally, the grammar needs improvements, use writing applications.

#6. The discussion section is poor and limited in justifying the similarity and differences with other studies using scientific facts. References (scientific facts) should be used to explain and back claims of significant association, similarities, and differences with other studies. The entirety of the discussion needs more reflective writing and synthesis.

#7. The citations need to be consistent all over, for instance, in the sentence, “This finding was supported by studies done in Ethiopia (31).” The ‘studies’ were not cited in the manuscript

#8. The final paragraph of the discussion deals with the Strengths and Limitations of the study, which I would prefer as an independent section but it is perfectly fine to include under discussion also.

#9. The Acronym and abbreviation section is not consistent with the document, limit to those used three or more times in the manuscript, have you even used RR?. Are they abbreviations or acronyms?

#10. Rewrite the acknowledgment so that it could make grammatical sense, Measure DHS?

Also, please ensure that, before resubmission, a person proficient in written English edits the manuscript. It is important that the message being conveyed in the manuscript is as unambiguous as possible

Reviewer #2: The study aims to assess the individual and community-level factors associated with Skilled Birth Attendant (SBA) delivery in Ethiopia.

Comments

Abstracts

The full name for ANC is to be stated prior to the use of the abbreviation ANC.

Method

Statement 'Finally, AOR with 95% CI and random effects were reported' requires revision.

Results

For 'wealth index(AOR=0.64;95% CI 0.46-0.87)' the figures to be spaced out.

Introduction

Paragraph 1, skilled birth attended. The words attendance or attended or attendants are to be standardized where necessary throughout the manuscript.

Paragraph 3, typo error ‘SBAdelivery’. The subsequent words ‘skilled birth attendant’ is to be replaced with SBA.

Independent variables

Typo error ‘visit(No visit’

Multi level logistic regression analysis

Typo error ‘ regression model Variables which’

Statement ‘Bivariable multilevel logistic regression analysis were considered for the individual and community level model adjustments for the multivariable multilevel logistic regression model’ requires revision.

Typo error ‘log likely hood’, ‘highest log likely hood ratio’

Statement ‘Finally, the adjusted odds ratio with a 95% confidence interval was reported for statistically significant variables.’ requires revision.

A statement on multicollinearity assessment (if any) to be included.

Results

Random effect analysis results

Technically p value cannot be zero (to use symbol p< )

Fixed effect analysis results

For the statement ‘Women who started ANC visit first trimester and second trimester had 1.87(AOR=1.87:95%CI 1.09-3.20) and 2.65 (AOR=2.65;95% CI 1.52-4.65)’ the figures 1.87(AOR=1.87:95%CI 1.09-3.20) and 2.65 (AOR=2.65;95% CI 1.52-4.65) are to be referred as second and first trimester respectively in the text.

AIC and BIC are to be included as part of the model fit diagnostic.

Discussion

References to be spaced out e.g. East Africa(10), Ghana(14), and East Africa(10).

Conclusion

Typo ‘It is Better to’

Citation of references in the text to follow journal format e.g. ( ) to be replaced with [ ]. Not all references in the list of references are conformed to the journal format.

6. PLOS authors have the option to publish the peer review history of their article (what does this mean?). If published, this will include your full peer review and any attached files.

Reviewer #1: **Yes: **Bikila Tefera Debelo

Reviewer #2: No

---

## [Author Response · Author response to Decision Letter 0]

2 Feb 2023

Rebuttal letter Date February 2, 2023

Subject; submission of revised manuscript (PONE-D-22-07743)

Individual and Community-level Determinants of Skilled Birth Attendant Delivery in Ethiopia; Multilevel Analysis

Hiwotie Getaneh Ayalew

To PLOS ONE

Dear all,

We would like to thank you for these constructive, building, and improvable comments on this manuscript that would improve the substance and content of the manuscript. We considered each comment and clarification question of editors and reviewers on the manuscript thoroughly. Our point-by-point responses for each comment and question are described in detail on the following pages. Further, the details of changes were shown by track changes in the supplementary document attached. The manuscript language was further improved in the revised manuscript. and we follow journal guidelines. We have also revised the statistical analysis method again and we have attached recent comments in a point-by-point response.

Version 1; editor’s comments

Authors’ response; thank you dear editor we have prepared the documents based on PLOS ONE requirements.

2. Thank you for stating the following financial disclosure: “no”

Authors’ response; thanks, dear editor. We have addressed this comment by stating that “The authors received no specific funding for this work.”

3. Thank you for stating the following in your Competing Interests section. “NO” Please complete your Competing Interests on the online submission form to state any Competing Interests. If you have no competing interests, please state "The authors have declared that no competing interests exist.", as detailed online in our guide for authors at http://journals.plos.org/plosone/s/submit-now

Authors’ response; thanks, dear editor. We have explained the competing interest as “The authors have declared that no competing interests exist” both in the manuscript and in the online submission section.

Version 2; reviewers’ comments

1. Is the manuscript technically sound, and do the data support the conclusions?

Reviewer #1: Partly

Reviewer #2: Yes

Authors’ response; thank you, dear reviewers. We have improved the revised version of the manuscript.

2. Has the statistical analysis been performed appropriately and rigorously?

Reviewer #1: Yes

Reviewer #2: Yes

Authors’ response; thank you, dear reviewers. 

3. Have the authors made all data underlying the findings in their manuscript fully available?

Reviewer #1: Yes

Reviewer #2: Yes

Authors’ response; Thank you, dear reviewers. We have stated that the data was fully available without restriction in the revised manuscript at the declaration session and online submission.

4. Is the manuscript presented in an intelligible fashion and written in Standard English?

Reviewer #1: No

Reviewer #2: Yes

Authors’ response; thanks a lot, dear reviewer. We have critically improved the readability of the manuscript. Please see the revised version.

Reviewer 1 Comments 

 Comments #1. There are numerous typographical and grammatical problems throughout the document which need thorough revision.

Authors’ response; Thanks a lot dear reviewer for your critical comment to improve the manuscript. We have fully accepted your comment and we have revised the manuscript to reduce the typological and grammatical errors. 

Comment #2. In the Introduction, additional justification is needed in the introduction section to justify the novelty of the study, “Even though skilled birth attendant delivery depends on both individual and community-level determinants, still, limited studies have been done beyond individual-level factors” is not enough of a justification since numerous significant factors this study reported had also been reported by other similar studies (which are used in the present study as a reference also)

Authors’ response; thanks a lot dear reviewer for your unlimited effort to improve the manuscript. We accepted your comment and included sound justification in the revised manuscript. 

Comment #3. In the Method section, It is important to specify whether the independent variables (individual and community) were taken as they are or were classified by the authors, for instance, was PCA conducted for the wealth Index?

Authors’ response; thanks, dear reviewer for your detailed revision. We have used some variables as coded as DHS data set and some variables which do not fulfill the chi-square assumption were recoded again based on some previous studies. Regarding the wealth index, PCA was conducted by the DHS program data analyzer. Dear reviewer, we kindly request you to see the DHS recode manual guide, which further expressed, how PCA was done for the wealth index. 

Comment #4. It is also important to include the measurement section in the Method for some of the variables especially for Community poverty, Community media exposure---

Authors’ response; Thanks a lot, dear reviewer. We have expressed in the revised manuscript, how these community-level factors are measured. we kindly request you see the revised manuscript.

Comment #5. The writing especially in the result section could be improved, what is written is stand-alone sentences, try connecting using conjunctions during interpretations. Additionally, the grammar needs improvements, use writing applications.

Authors’ response; Thank you in advance dear reviewer. We have tried to revise the result section in the revised manuscript. We have also used Grammarly and expert persons in English for the improvement of the revised manuscript.

Comment #6. The discussion section is poor and limited in justifying the similarity and differences with other studies using scientific facts. References (scientific facts) should be used to explain and back claims of significant association, similarities, and differences with other studies. The entirety of the discussion needs more reflective writing and synthesis

Authors’ response; Thank you dear reviewer for your great effort to review our manuscript. We have revised the discussion again and added some scientific evidence with their references in the revised manuscript.

Comment #7. The citations need to be consistent all over, for instance, in the sentence, “This finding was supported by studies done in Ethiopia (31).” The ‘studies’ were not cited in the manuscript 

Authors’ response; Thanks dear reviewer we have appreciated your comments. We have revised the typing error. we kindly request you see it in the revised manuscript.

Comment #8. The final paragraph of the discussion deals with the Strengths and Limitations of the study, which I would prefer as an independent section but it is perfectly fine to include under discussion also.

Authors’ response; Thanks a lot dear reviewer for your deep comments.

Comment #9. The Acronym and abbreviation section is not consistent with the document, limit to those used three or more times in the manuscript, have you even used RR?. Are they abbreviations or acronyms?

Authors’ response; thank you, dear reviewer, for your unlimited effort. The abbreviation RR was not used and we remove it in the revised manuscript.

Comment #10. Rewrite the acknowledgment so that it could make grammatical sense, Measure DHS? Also, please ensure that, before resubmission, a person proficient in written English edits the manuscript. It is important that the message being conveyed in the manuscript is as unambiguous as possible

Authors’ response; thanks dear reviewer for your constructive comments. We have revised and rewritten the acknowledgment in the revised manuscript. Dear reviewer a person proficient in the English language revised and edits the revised manuscript before submission.

Reviewer #2 comments

The study aims to assess the individual and community-level factors associated with Skilled Birth Attendant (SBA) delivery in Ethiopia.

Comment #1. Abstracts 

The full name for ANC is to be stated prior to the use of the abbreviation ANC 

Authors’ response; thanks, dear reviewer. We have accepted your comment. We have stated ANC before using it as an abbreviation in the revised manuscript.

Comment #2. Method 

Statement 'Finally, AOR with 95% CI and random effects were reported' requires revision..

Authors’ response; thank your dear reviewer for your contractive comment

Comment# 3. Results 

For 'wealth index(AOR=0.64;95% CI 0.46-0.87)' the figures to be spaced out 

Authors’ response; thank you, dear reviewer. we have edited the revised manuscript.

 Comment #4. Introduction

Paragraph 1, skilled birth attended. The words attendance or attended or attendants are to be standardized where necessary throughout the manuscript.

Paragraph 3, typo error ‘SBAdelivery’. The subsequent words ‘skilled birth attendant’ is to be replaced with SBA.

Authors’ response; thank you, dear reviewer. We strongly appreciate your comment. We have revised the introduction in the revised manuscript and we take consistent words for the skilled birth attendant. We also replaced skilled birth attendant delivery with SBA delivery.

 Comment #5. Independent variables

Typo error ‘visit(No visit’

Authors’ response; Thanks a lot, dear reviewer. We have edited the type error.

Comment #6. Multilevel logistic regression analysis

Typo error ‘ regression model Variables which’

Statement ‘Bivariable multilevel logistic regression analysis were considered for the individual and community level model adjustments for the multivariable multilevel logistic regression model’ requires revision.

Typo error ‘log likely hood’, ‘highest log likely hood ratio’

Statement ‘Finally, the adjusted odds ratio with a 95% confidence interval was reported for statistically significant variables.’ requires revision.

A statement on multicollinearity assessment (if any) to be included.

Authors’ response; thanks dear reviewer we appreciate your comments. We have edited the suggested typing errors in the revised manuscript and we have revised the statements again. Regarding multicollinearity assessment,we have again revised data set extracted for analysis and cheked multicolinerity for indipendant variables. Those variables with VIF less than 10 were taken to fit the model.

Comment #7.Results 

Random effect analysis results, Technically p value cannot be zero (to use symbol p<)

Fixed effect analysis results, 

For the statement ‘Women who started ANC visit first trimester and second trimester had 1.87(AOR=1.87:95%CI 1.09-3.20) and 2.65 (AOR=2.65;95% CI 1.52-4.65)’ the figures 1.87(AOR=1.87:95%CI 1.09-3.20) and 2.65 (AOR=2.65;95% CI 1.52-4.65) are to be referred as second and first trimester respectively in the text.

 AIC and BIC are to be included as part of the model fit diagnostic.

Authors’ response; thanks a lot dear reviewer we have critically improved typing and technical errors. Regarding model comparison for nested model as you know devieance and loglikelihood ratio are commonly used and recommended since they test a hypothesis that weather additional parameter has improved model fit or not. Whereas AIC and BIC are commonly used to compare models from different families with lowest value being best. Therefore, in our analysis multilevel model is hierarchical and models are nested (Singer J, Willett J, Singer J, Willett J. Doing data analysis with the multilevel model for change. Applied longitudinal data analysis: Modeling change and event occurrence. 2003:96-7.). On the other hand,Akaike Information Criterion (Akaike,1974) and Bayesian Information Criterion (schwarz,1978) are a measure of goodness of fit for model summaries. These are penalized for many parameters. A smaller value is better, but no absolute good value, whereas loglikelihood hood and deviance are unpenalized goodness of fit. Always negative, larger (close to zero) is better .deviance is -2log likelihood and the smaller is better. Hence we preferred to use log-likelihood and deviance for the model fitness measure . 

Comment#8 Discussion

References to be spaced out e.g. East Africa(10), Ghana(14), and East Africa(10).

Authors’ response; thanks, dear reviewer. We have edited the typing error.

Comment #9 Conclusion

Typo ‘It is Better to

Authors’ response; thanks in advance, dear reviewer

Thanks, a lot!!!

---

## [Editor Report · Decision Letter 1]

5 Feb 2023

PONE-D-22-07743R1Individual and community-level determinants of skilled birth attendant delivery in Ethiopia; multilevel analysisPLOS ONE

Dear Dr. Ayalew,

Thank you for submitting your manuscript to PLOS ONE. After careful consideration, we feel that it has merit but does not fully meet PLOS ONE’s publication criteria as it currently stands. Therefore, we invite you to submit a revised version of the manuscript that addresses the points raised during the review process.

We look forward to receiving your revised manuscript.

Kind regards,

Seifadin Ahmed Shallo, MPH

Academic Editor

PLOS ONE

Additional Editor Comments (if provided):

Dear authors, I appreciate your responses you gave on the comments given for you. However, there are some issues to be addressed yet

1. what value will this paper add to the existing science on the matter (novelty of the finding). The findings you are reporting are those already identified and well known. Even you have missed some community/individual level factors such as women status/decision making power from your analysis. Don't you think your finding is incomplete/biased because of missing some main important variables.

2. Under your recommendation (in both abstract and recommendation section) you have recommended, increasing the timely initiation of ANC through health education. But, what evidence do you have as the gap is the gap of knowledge among the women?? how do you justify your recommendation from the perspective of what the government is currently doing?

3. In addition, you have recommended HE should focus on low income women. but, your recommendation is not supported with evidence. plus, improving living standard of the women. how much applicable this recommendation? And the recommendation is too general?. its better to make more specific recommendation and applicable

4. are HEWs considered as SBA?? pls present your evidence(reference)

5. what does community level media exposure mean?? is it really community level factor or individual level? better to have objective reference?

6. you have explained as the community level factors were grouped based on median because of the skewness of the data. the question here is since the community level variables are categorical, how did you check for normality??? is it possible?

7. how did you manage confounding issue? e.g. individual level economic status and community level wealth index are same thing measured in different approaches? so, don't you think this could create collinearity?

8. why you measure wealth index at individual and community level with different scale? i.e you have said poor, middle and rich for individual and only two (i didn't understood what are they) at community level. and also u have used median for classification for community level.

9. the standard wealth index is classification is 5. in your case you have said 3 for individual and 2 for community level. what is your justification for this? and could you able to compare with data analyzed on 5 scale based classification.
---

## [Author Response · Author response to Decision Letter 1]

11 Mar 2023

Rebuttal letter Date, March 8, 2023

Subject; submission of revised manuscript (PONE-D-22-07743)

Individual and Community-level Determinants of Skilled Birth Attendant Delivery in Ethiopia; Multilevel Analysis

Hiwotie Getaneh Ayalew

To PLOS ONE

Dear all,

We would like to thank you for these constructive, building, and improvable comments on this manuscript that would improve the substance and content of the manuscript. We considered each comment and clarification question of editors and reviewers on the manuscript thoroughly. Our point-by-point responses for each comment and question are described in detail on the following pages. Further, the details of changes were shown by track changes in the supplementary document attached. The manuscript language was further improved in the revised manuscript. and we follow journal guidelines. We have also revised the statistical analysis method again and we have attached recent comments in a point-by-point response.

Version 1; editor’s comments

Authors’ response; thank you dear editor we have prepared the documents based on PLOS ONE requirements.

2. Thank you for stating the following financial disclosure: “no”

Authors’ response; thanks, dear editor. We have addressed this comment by stating that “The authors received no specific funding for this work.”

3. Thank you for stating the following in your Competing Interests section. “NO” Please complete your Competing Interests on the online submission form to state any Competing Interests. If you have no competing interests, please state "The authors have declared that no competing interests exist.", as detailed online in our guide for authors at http://journals.plos.org/plosone/s/submit-now

Authors’ response; thanks, dear editor. We have explained the competing interest as “The authors have declared that no competing interests exist” both in the manuscript and in the online submission section.

Version 1.1; Additional editor’s comments

1. what value will this paper add to the existing science on the matter (novelty of the finding). The findings you are reporting are those already identified and well known. Even you have missed some community/individual level factors such as women status/decision making power from your analysis. Don't you think your finding is incomplete/biased because of missing some main important variables.

Authors’ response: thanks a lot dear editor for your unlimited effort to improve the manuscript. We accepted your comment and included sound justification in the revised manuscript. When we come to the variables, those missed variables in the analysis do not fulfill the chi-square assumption and the VIF was greater than 10. So that due to this reason, we exclude these variables in the analysis during model fitness.

2. Under your recommendation (in both abstract and recommendation section) you have recommended, increasing the timely initiation of ANC through health education. But, what evidence do you have as the gap is the gap of knowledge among the women?? how do you justify your recommendation from the perspective of what the government is currently doing?

Authors' response; thanks a lot dear editor for your constructive comments. Timely ANC visit was a factor that was significantly associated with SBA utilization in Ethiopia. From the perspective of the government concern, timely ANC visits will be improved if those pregnant mothers get health education at health institutions. because most mothers delayed from their ANC visit due to lack of knowledge. This is supported by different research pieces of evidence previously.

3. In addition, you have recommended HE should focus on low income women. but, your recommendation is not supported with evidence. plus, improving living standard of the women. how much applicable this recommendation? And the recommendation is too general?. its better to make more specific recommendation and applicable

Authors' response; thank you in advance, dear editor. We take your comments we try to make specific recommendations. Regarding pieces of evidence we have revised different research articles and the national health maternal health guideline standard to recommend our significant findings of the thesis.

4. are HEWs considered as SBA?? pls present your evidence(reference)

Authors' response; thanks dear editor for your constructive comments. HEWs are considered SBA. we have a piece of evidence that is written in the revised manuscript and also the EDHS data set considers the HEWs as SBA. We kindly request you see the DHS data set and the evidence we have added to the revised manuscript.

5. what does community level media exposure mean?? is it really community level factor or individual level? better to have objective reference?

Authors' response; thanks a lot dear editor for your comments. Media exposure was an individual-level factor whereas community-level media exposure was a community-level factor. The individual level factor was obtained by adding variables (households having radio and TV) whereas community-level media exposure, was obtained by aggregating the individual level media exposure in each cluster by using the proportion of those who had media exposure and this community-level media exposure shows the overall media exposure in the community. It was categorized as two groups based on median values of high and low because the aggregated variable had a skewed distribution.

6. you have explained as the community level factors were grouped based on median because of the skewness of the data. the question here is since the community level variables are categorical, how did you check for normality??? is it possible?

Authors response; thank you, dear editor. yes, it is possible. we have used a histogram to check the normality of the data for analysis.

7. how did you manage confounding issue? e.g. individual level economic status and community level wealth index are same thing measured in different approaches? so, don't you think this could create collinearity

Authors response; thanks, dear editor. The confounders were managed by checking their multicollinearity using their VIF value. There was no multicollinearity between individual-level economic status and community-level wealth status and their VIF value was below 10.

8. why you measure wealth index at individual and community level with different scale? i.e you have said poor, middle and rich for individual and only two (i didn't understood what are they) at community level. and also u have used median for classification for community level.

Authors' response; thanks a lot, dear Editor. wealth index was measured using the PCA (principal component analysis), which was done by the DHS data analyst personnel and it was categorized into five groups poorest, poor, middle, rich, and richest. In our analysis, we categorize the individual level factors as poor, middle, and rich. This is because of that when we take the five groups as poorest, poor, middle, rich, and richest groups directly, it did not fulfill the chi-square assumption, so we prefer to classify it as three groups based on different literature.

The community-level wealth index was aggregated from the individual-level wealth index. For analysis, it was categorized as two groups based on their distribution on the histogram. Their distribution was skewed and we used median value for classification.

9. the standard wealth index is classification is 5. in your case you have said 3 for individual and 2 for community level. what is your justification for this? and could you able to compare with data analyzed on 5 scale based classification.

Authors' response; thanks a lot dear editor for your constructive comments. yes really the standard wealth index was classified into two five groups, but in our study, we classified the individual level wealth index into three groups based on different scientific evidence because the standard DHS classification directly does not fulfill the chi-square assumption. Whereas the community-level wealth index was aggregated from individual-level wealth index variables and was classified into two categories based on previous studies.

Version 2; reviewers’ comments

1. Is the manuscript technically sound, and do the data support the conclusions?

Reviewer #1: Partly

Reviewer #2: Yes

Authors’ response; thank you, dear reviewers. We have improved the revised version of the manuscript.

2. Has the statistical analysis been performed appropriately and rigorously?

Reviewer #1: Yes

Reviewer #2: Yes

Authors’ response; thank you, dear reviewers. 

3. Have the authors made all data underlying the findings in their manuscript fully available?

Reviewer #1: Yes

Reviewer #2: Yes

Authors’ response; Thank you, dear reviewers. We have stated that the data was fully available without restriction in the revised manuscript at the declaration session and online submission.

4. Is the manuscript presented in an intelligible fashion and written in Standard English?

Reviewer #1: No

Reviewer #2: Yes

Authors’ response; thanks a lot, dear reviewer. We have critically improved the readability of the manuscript. Please see the revised version.

Reviewer 1 Comments 

 Comments #1. There are numerous typographical and grammatical problems throughout the document which need thorough revision.

Authors’ response; Thanks a lot dear reviewer for your critical comment to improve the manuscript. We have fully accepted your comment and we have revised the manuscript to reduce the typological and grammatical errors. 

Comment #2. In the Introduction, additional justification is needed in the introduction section to justify the novelty of the study, “Even though skilled birth attendant delivery depends on both individual and community-level determinants, still, limited studies have been done beyond individual-level factors” is not enough of a justification since numerous significant factors this study reported had also been reported by other similar studies (which are used in the present study as a reference also)

Authors’ response; thanks a lot dear reviewer for your unlimited effort to improve the manuscript. We accepted your comment and included sound justification in the revised manuscript. 

Comment #3. In the Method section, It is important to specify whether the independent variables (individual and community) were taken as they are or were classified by the authors, for instance, was PCA conducted for the wealth Index?

Authors’ response; thanks, dear reviewer for your detailed revision. We have used some variables as coded as DHS data set and some variables which do not fulfill the chi-square assumption were recoded again based on some previous studies. Regarding the wealth index, PCA was conducted by the DHS program data analyzer. Dear reviewer, we kindly request you to see the DHS recode manual guide, which further expressed, how PCA was done for the wealth index. 

Comment #4. It is also important to include the measurement section in the Method for some of the variables especially for Community poverty, Community media exposure---

Authors’ response; Thanks a lot, dear reviewer. We have expressed in the revised manuscript, how these community-level factors are measured. we kindly request you see the revised manuscript.

Comment #5. The writing especially in the result section could be improved, what is written is stand-alone sentences, try connecting using conjunctions during interpretations. Additionally, the grammar needs improvements, use writing applications.

Authors’ response; Thank you in advance dear reviewer. We have tried to revise the result section in the revised manuscript. We have also used Grammarly and expert persons in English for the improvement of the revised manuscript.

Comment #6. The discussion section is poor and limited in justifying the similarity and differences with other studies using scientific facts. References (scientific facts) should be used to explain and back claims of significant association, similarities, and differences with other studies. The entirety of the discussion needs more reflective writing and synthesis

Authors’ response; Thank you dear reviewer for your great effort to review our manuscript. We have revised the discussion again and added some scientific evidence with their references in the revised manuscript.

Comment #7. The citations need to be consistent all over, for instance, in the sentence, “This finding was supported by studies done in Ethiopia (31).” The ‘studies’ were not cited in the manuscript 

Authors’ response; Thanks dear reviewer we have appreciated your comments. We have revised the typing error. we kindly request you see it in the revised manuscript.

Comment #8. The final paragraph of the discussion deals with the Strengths and Limitations of the study, which I would prefer as an independent section but it is perfectly fine to include under discussion also.

Authors’ response; Thanks a lot dear reviewer for your deep comments.

Comment #9. The Acronym and abbreviation section is not consistent with the document, limit to those used three or more times in the manuscript, have you even used RR?. Are they abbreviations or acronyms?

Authors’ response; thank you, dear reviewer, for your unlimited effort. The abbreviation RR was not used and we remove it in the revised manuscript.

Comment #10. Rewrite the acknowledgment so that it could make grammatical sense, Measure DHS? Also, please ensure that, before resubmission, a person proficient in written English edits the manuscript. It is important that the message being conveyed in the manuscript is as unambiguous as possible

Authors’ response; thanks dear reviewer for your constructive comments. We have revised and rewritten the acknowledgment in the revised manuscript. Dear reviewer a person proficient in the English language revised and edits the revised manuscript before submission.

Reviewer #2 comments

The study aims to assess the individual and community-level factors associated with Skilled Birth Attendant (SBA) delivery in Ethiopia.

Comment #1. Abstracts 

The full name for ANC is to be stated prior to the use of the abbreviation ANC 

Authors’ response; thanks, dear reviewer. We have accepted your comment. We have stated ANC before using it as an abbreviation in the revised manuscript.

Comment #2. Method 

Statement 'Finally, AOR with 95% CI and random effects were reported' requires revision..

Authors’ response; thank your dear reviewer for your contractive comment

Comment# 3. Results 

For 'wealth index(AOR=0.64;95% CI 0.46-0.87)' the figures to be spaced out 

Authors’ response; thank you, dear reviewer. we have edited the revised manuscript.

 Comment #4. Introduction

Paragraph 1, skilled birth attended. The words attendance or attended or attendants are to be standardized where necessary throughout the manuscript.

Paragraph 3, typo error ‘SBAdelivery’. The subsequent words ‘skilled birth attendant’ is to be replaced with SBA.

Authors’ response; thank you, dear reviewer. We strongly appreciate your comment. We have revised the introduction in the revised manuscript and we take consistent words for the skilled birth attendant. We also replaced skilled birth attendant delivery with SBA delivery.

 Comment #5. Independent variables

Typo error ‘visit(No visit’

Authors’ response; Thanks a lot, dear reviewer. We have edited the type error.

Comment #6. Multilevel logistic regression analysis

Typo error ‘ regression model Variables which’

Statement ‘Bivariable multilevel logistic regression analysis were considered for the individual and community level model adjustments for the multivariable multilevel logistic regression model’ requires revision.

Typo error ‘log likely hood’, ‘highest log likely hood ratio’

Statement ‘Finally, the adjusted odds ratio with a 95% confidence interval was reported for statistically significant variables.’ requires revision.

A statement on multicollinearity assessment (if any) to be included.

Authors’ response; thanks dear reviewer we appreciate your comments. We have edited the suggested typing errors in the revised manuscript and we have revised the statements again. Regarding multicollinearity assessment,we have again revised data set extracted for analysis and cheked multicolinerity for indipendant variables. Those variables with VIF less than 10 were taken to fit the model.

Comment #7.Results 

Random effect analysis results, Technically p value cannot be zero (to use symbol p<)

Fixed effect analysis results, 

For the statement ‘Women who started ANC visit first trimester and second trimester had 1.87(AOR=1.87:95%CI 1.09-3.20) and 2.65 (AOR=2.65;95% CI 1.52-4.65)’ the figures 1.87(AOR=1.87:95%CI 1.09-3.20) and 2.65 (AOR=2.65;95% CI 1.52-4.65) are to be referred as second and first trimester respectively in the text.

 AIC and BIC are to be included as part of the model fit diagnostic.

Authors’ response; thanks a lot dear reviewer we have critically improved typing and technical errors. Regarding model comparison for nested model as you know devieance and loglikelihood ratio are commonly used and recommended since they test a hypothesis that weather additional parameter has improved model fit or not. Whereas AIC and BIC are commonly used to compare models from different families with lowest value being best. Therefore, in our analysis multilevel model is hierarchical and models are nested (Singer J, Willett J, Singer J, Willett J. Doing data analysis with the multilevel model for change. Applied longitudinal data analysis: Modeling change and event occurrence. 2003:96-7.). On the other hand,Akaike Information Criterion (Akaike,1974) and Bayesian Information Criterion (schwarz,1978) are a measure of goodness of fit for model summaries. These are penalized for many parameters. A smaller value is better, but no absolute good value, whereas loglikelihood hood and deviance are unpenalized goodness of fit. Always negative, larger (close to zero) is better .deviance is -2log likelihood and the smaller is better. Hence we preferred to use log-likelihood and deviance for the model fitness measure . 

Comment#8 Discussion

References to be spaced out e.g. East Africa(10), Ghana(14), and East Africa(10).

Authors’ response; thanks, dear reviewer. We have edited the typing error.

Comment #9 Conclusion

Typo ‘It is Better to

Authors’ response; thanks in advance, dear reviewer

Thanks, a lot!!!

---

## [Editor Report · Decision Letter 2]

13 Apr 2023

PONE-D-22-07743R2Individual and community-level determinants of skilled birth attendant delivery in Ethiopia; multilevel analysisPLOS ONE

Dear Dr. Ayalew,

Thank you for submitting your manuscript to PLOS ONE. After careful consideration, we feel that it has merit but does not fully meet PLOS ONE’s publication criteria as it currently stands. Therefore, we invite you to submit a revised version of the manuscript that addresses the points raised during the review process.

We look forward to receiving your revised manuscript.

Kind regards,

Seifadin Ahmed Shallo, MPH

Academic Editor

PLOS ONE

Journal Requirements:

**Additional Editor Comments:**

Dear authors, I highly appreciate your response and explanations you did to the comments raised on your manuscript!. However, some of the responses you gave for the questions are not scientifically convincing.

For instance

1. Under your recommendation (in both abstract and recommendation section) you have recommended, increasing the timely initiation of ANC through health education. But, what evidence do you have as the gap is the gap of knowledge among the women?? how do you justify your recommendation from the perspective of what the government is currently doing?

your response is not inline with the question. Recommendation should be inline with your finding. But, what you present as an evidence is already existing knowledge, not your finding

2. you have explained as the community level factors were grouped based on median because of the skewness of the data. the question here is since the community level variables are categorical, how did you check for normality??? is it possible?

Authors response; thank you, dear editor. yes, it is possible. we have used a histogram to check the normality of the data for analysis.

its miracle for me to use histogram for categorical data, which is totally impossible. please, read biostatistics on data presentation

its hard to accept your manuscript as its now, unless you either justify or modify the above comments/questions

---

## [Author Response · Author response to Decision Letter 2]

11 Jul 2023

Rebuttal letter May 17, 2023

Subject; submission of revised manuscript (PONE-D-22-07743)

INDIVIDUAL AND COMMUNITY-LEVEL DETERMINANTS OF SKILLED BIRTH ATTENDANT DELIVERY IN ETHIOPIA; MULTILEVEL ANALYSIS

Hiwotie Getaneh Ayalew

To PLOS ONE

Dear all,

We would like to thank you for these constructive, building, and improvable comments on this manuscript that would improve the substance and content of the manuscript. We considered each comment and clarification question of editors and reviewers on the manuscript thoroughly. Our point-by-point responses for each comment and question are described in detail on the following pages. Further, the details of changes were shown by track changes in the supplementary document attached. 

editor’s comments

Dear Dr. Ayalew,

Authors’ response: many thanks dear reviewer for giving me additional opportunity to revise the manuscript. I have checked all the references and they are correct and complete. No reteracted articles are cited.

Comment 1: Under your recommendation (in both abstract and recommendation section) you have recommended, increasing the timely initiation of ANC through health education. But, what evidence do you have as the gap is the gap of knowledge among the women?? how do you justify your recommendation from the perspective of what the government is currently doing?

your response is not inline with the question. Recommendation should be inline with your finding. But, what you present as an evidence is already existing knowledge, not your finding

Authors’ response: many thanks dear editor. We are much convinced with your suggestion and we revised the recommendation section in the abstract and main text (kindly refer too revised manuscript.

Comment 2: you have explained as the community level factors were grouped based on median because of the skewness of the data. the question here is since the community level variables are categorical, how did you check for normality??? is it possible? Authors response; thank you, dear editor. yes, it is possible. we have used a histogram to check the normality of the data for analysis. its miracle for me to use histogram for categorical data, which is totally impossible. please, read biostatistics on data presentation

Authors’ response: thank you so much dear reviewer. Sorry for creating confusion. Ofcurse as you said it is imposible to check normality by using histogram for categorical variable. Let me explain again how we come across community level variables. Initially these ggregated community level variables are proportions not a categorical variables. For instance, community media exposure is a proportion computed by dividing number of individuals exposed to media to total population in that cluster. As you know proportion is continuous variable and to categorize as high and low media exposure level we need to use either mean (if the proportion is normally distributed) or median (if the proportion is skewed). To decide weither mean or mrdian should be used for categorizing proportion of media exposure we ckecked normality by using histogram and skewed distribution was observed. Then we used median for categorization. It is therefore, for the proportion that histogram is used not for categorized variable. A similar approach was used for community poverty level and we have clearly stated these in the method section and we kindly refer to pragraph six of method section. We believe we have clarified it now.

Many thanks for your commitment to improve the quality of our manuscript.

---

## [Editor Report · Decision Letter 3]

13 Jul 2023

Individual and community-level determinants of skilled birth attendant delivery in Ethiopia; multilevel analysis

PONE-D-22-07743R3

Dear Dr. Ayalew,

We’re pleased to inform you that your manuscript has been judged scientifically suitable for publication and will be formally accepted for publication once it meets all outstanding technical requirements.

Kind regards,

Seifadin Ahmed Shallo, MPH

Academic Editor

PLOS ONE
---

## [Editor Report · Acceptance letter]

21 Jul 2023

PONE-D-22-07743R3 

Individual and community-level determinants of skilled birth attendant delivery in Ethiopia; multilevel analysis 

Dear Dr. Ayalew:

I'm pleased to inform you that your manuscript has been deemed suitable for publication in PLOS ONE. Congratulations! Your manuscript is now with our production department. 

Kind regards, 

on behalf of

Prof. Seifadin Ahmed Shallo 

Academic Editor

PLOS ONE